# Wavelength and Light Intensity Affect Macro- and Micronutrient Uptake, Stomata Number, and Plant Morphology of Common Bean (*Phaseolus vulgaris* L.)

**DOI:** 10.3390/plants13030441

**Published:** 2024-02-02

**Authors:** Paulo Mauricio Centenaro Bueno, Wagner A. Vendrame

**Affiliations:** 1Instituto Federal do Paraná, Campus Palmas, Palmas 85555-000, PR, Brazil; 2Environmental Horticulture Department, Institute of Food and Agricultural Sciences, University of Florida, Gainesville, FL 32611, USA; vendrame@ufl.edu

**Keywords:** photomorphogenesis, LED light quality, plant factory, nutritional quality

## Abstract

It is already known that light quality and intensity have major influences on the growth, etiolation, germination, and morphology of many plant species, but there is limited information about the effect of wavelength and light intensity on nutrient absorption by plants. Therefore, this study was established to evaluate the plant growth, stomata formation, chlorophyll index, and absorption of macro- and micronutrients by common bean plants under six light treatments. The experimental design was completely randomized and consisted of six treatments: strong blue (blue LED at high light intensity); weak blue (blue LED at low light intensity); strong red (red LED at high light intensity); weak red (red LED at low light intensity; pink (combined red + blue LED), and white (combined red + white led). The stomatal density (stomata mm^−2^); the SPAD index; plant height (cm); root length (cm); plant dry weight (g); root dry weight (g); and the concentrations of N, S, K, Mg, Ca, B, Zn, Mn, and Fe on leaf analysis were influenced by all treatments. We found that plant photomorphogenesis is controlled not only by the wavelength, but also by the light intensity. Etiolation was observed in bean plants under blue light at low intensity, but when the same wavelength had more intensity, the etiolation did not happen, and the plant height was the same as plants under multichromatic lights (pink and white light). The smallest plants showed the largest roots, some of the highest chlorophyll contents, and some of the highest stomatal densities, and consequently, the highest dry weight, under white LED, showing that the multichromatic light at high intensity resulted in better conditions for the plants in carbon fixation. The effect of blue light on plant morphology is intensity-dependent. Plants under multichromatic light tend to have lower concentrations of N, K, Mg, and Cu in their leaves, but the final amount of these nutrients absorbed is higher because of the higher dry weight of these plants. Plants under blue light at high intensity tended to have lower concentrations of N, Cu, B, and Zn when compared to the same wavelength at low intensity, and their dry weight was not different from plants grown under pink light. New studies are needed to understand how and on what occasions intense blue light can replace red light in plant physiology.

## 1. Introduction

It is already known that light quality, intensity, and photoperiod can affect plant behavior [1,2]. Many studies have been established to determine the most efficient sources of light that provide good plant growth and are sustainable, mainly using light emitting diodes (LEDs), which are more efficient and cost-effective [3] and which have also been widely used in microgravity studies aiming to develop space life support systems [4]. These sources of light have been used in vertical farms, greenhouses [4,5,6,7], and even in research on field crops [5].

Other advantages of LEDs, compared to fluorescent bulbs, include their truly low heat generation, smaller size, and longer life, and for this kind of study, the selective light intensity and quality [8,9], which allows the control of photosynthetically active radiation (PAR) [10]. PAR corresponds to the wavelength between 400 (blue light) and 700 nanometers (red light), and it is absorbed by the photosynthetic pigments (chlorophyll *a* and *b*). There are also other important pigments, such as cryptochromes and phytochromes, that detect blue and red light, respectively [11].

Light quality, intensity, and photoperiod regulate a plant’s growth and its development throughout its life cycle [12,13]. The growth and the plant are dependent on PAR, important for photosynthesis, photomorphogenesis, and phototropism [14]. The plant’s net photosynthetic rate is affected by the amount of light received per plant, so plant morphology can increase the amount of light received by plants and their growth rates. Plant morphology is affected by spectral distribution via multiple photoreceptors, and it is also affected by photon flux density; therefore, there may be interaction between these effects. It is important to understand these effects and interactions to control plant growth and morphology using LEDs [15]. Light quality can even affect plant yield and its nutritional quality [16,17].

Blue light can suppress plant elongation via the cryptochrome reaction, but on the other hand, sole blue light decreases the phytochrome photostationary state (PSS) and has an elongation promotion effect via the phytochrome reaction. However, blue light has a weak effect on phytochromes, and it is canceled out by red light with comparable photosynthetic photon flux density (PPFD) [15]. Sole blue light elongated cucumber stems [18], possibly due to the action of phytochromes [15].

Red light is absorbed by phytochromes and suppresses plant elongation. Lighting with combined blue and red light suppresses plant elongation more than that caused by lighting with sole blue and sole red light [19], because elongation is suppressed by the actions of both the cryptochrome and phytochrome reactions [15].

Far-red light is absorbed by phytochromes and promotes plant elongation [15], and adding far-red light can promote uniformly sized seedlings [20]. It not only depends on light quality, but on its intensity too [21], so plants grown under weak light tend to etiolate, and plants grown under intense light show compact shapes [15]. With increasing sucrose concentrations in *Arabidopsis thaliana*, leaf blade expansion was inhibited in white light but was enhanced in darkness [22]. Light quality and intensity work together in plant elongation. Even under intense light with a PPFD of 300 μmol m^−2^ s^−1^, which naturally reduces growth in lettuce, making it too compact, a recent study showed that plants grew into normal shapes with sole blue light that promoted plant elongation [15].

Another study on the influence of radiation quality on the in vitro rooting and nutrient concentrations of peach rootstock showed that in roots, the concentrations of P, Ca, and Mg were constant in all treatments, while that of K decreased with dark, red, and green light. The Fe concentration in roots under green light was almost four times greater than that of white light. The P concentration in shoots increased in the dark treatment, more than that in the control (white). K and Ca concentrations did not differ in all the treatments, while the Mg concentration decreased under the dark treatment. The Fe concentration was twice as large as that in the control with the red and green treatment. Red light also increased the concentration of Zn, while that of Mn did not differ between the treatments [14]. An induced growth increase was observed with blue LEDs on lettuce leaves, and it was seen that high-intensity blue LEDs promote plant growth by controlling the integrity of chloroplast proteins that optimize photosynthetic performance in the natural environment [23].

The adequate combination of R (red light) and B (blue light) can improve plant growth and photosynthesis when compared with monochromatic light. In a recent study, R and B together were imperative for soybean chlorophyll biosynthesis and function. Sole red light could be adverse for soybean growth; it stimulates excessive carbohydrate accumulation in leaves during the daytime and disturbs the nocturnal growth of soybean. Sole blue light may be insufficient for soybean growth and development [5]. According to the same study, soybean plants grown with combinations of 80B:20R% and 50B:50R% exhibited suitable plant architecture for the controlled environment, showed higher CO_2_ assimilation rates and stomatal conductance, and accumulated higher biomass.

Understanding the effects and interactions between light quality and intensity is required to control plant growth and morphology using LEDs [15]. Therefore, the main objective of this study was to evaluate the growth, development, number of stomata, relative chlorophyll content and nutrient concentration of common bean (*Phaseolus vulgaris* L.) leaves under different light sources.

## 2. Results

The stomatal density; SPAD index; plant height; root length; shoot dry weight; root dry weight; and the concentrations of N, S, K, Mg, Ca, B, Zn, Mn, and Fe were influenced by the treatments, as shown in Figure 1, Figure 2, Figure 3 and Figure 4.

The relative chlorophyll content (SPAD) was higher (24.37) in plants under strong blue light, and the lowest level was found in plants under weak red light (18.75), which did not differ from strong red and weak blue light (20.78 and 19.96, respectively).

Etiolated plants were found under weak blue light (33 cm), while the smallest were found under white light (13 cm), which did not differ from strong red, strong blue, and pink light.

The smallest plants under the white light were the plants that showed the largest roots (28 cm), and the shortest were found in plants under strong blue (11.8 cm) and strong red light (15.63).

Plants under pink light showed the highest stomatal density (227 stomata mm^−2^), which did not differ from strong blue (175) and white light (187). On the other hand, plants under strong red, weak red, and weak blue light had the lowest stomatal densities in their leaves (92, 122, and 90, respectively), as can be seen in Figure 3. 

Regarding the concentration of nutrients in the leaves, plants under weak blue light had the highest concentrations (8.2%) of nitrogen and copper (16 ppm), while plants under white and pink light had the lowest concentrations of N (5.26% and 5.46%, respectively), and those under white, pink, and strong red light had the lowest concentrations of Cu (8, 10.7 and 10.7, respectively, Figure 3). This is probably because these treatments resulted in the highest amount of dry weight (Figure 5), and therefore, we estimated the total amount of each macro- and micronutrient absorbed by the plants.

The lowest concentration of S was found in plants under strong red light (0.35%), while the other treatments did not differ from each other. Plants under sole red and blue lights, independently of the intensity, had the highest levels of K, from 5.87% to 6.22%, while plants under white and pink light had the lowest levels of K (5.14 and 5.31, respectively).

Plants under strong red light had the highest concentrations of Mg in their leaves (0.58%), while the lowest concentrations were found in plants under white light (0.43%), which did not differ from pink light (0.49%).

The highest levels of Ca were found in plants under strong red (1.99%), weak red (1.96%), and pink light (1.92%), while the other treatments had concentrations ranging from 1.47 to 1.56%.

The concentrations of B (Figure 4) in plants were also influenced by the light treatments. The highest levels of B were found in plants under weak red (59 ppm), weak blue (59.67 ppm), and pink light (52 ppm). This last treatment did not differ from that of strong blue and white light (47.33 ppm and 43.33 ppm, respectively).

The lowest levels of Zn were found in plants under strong red, strong blue, and white light (57, 54, and 54, ppm, respectively), while the highest levels were found in plants under weak red, weak blue, and pink light (70.3, 70.7, and 81.7, respectively).

There was a significant difference between the lowest (175 ppm under white light) and the highest level of Mn (283.7 ppm under pink light). This last treatment did not differ from that of strong red, weak red, and weak blue light. 

The highest concentration of Fe was found in plants under pink light (175.3 ppm) when compared to all other treatments, but they did not differ from each other, with concentrations ranging between 109 and 136 ppm.

The concentration of Cu in plants under weak blue light was twice (16 ppm) that found in plants under white light (8 ppm).

The averages of shoot dry weight, root dry weight, and total dry weight can be seen in Figure 5. The lowest averages of total dry weight were found under the lower-intensity lights (weak red, 0.25 g, and weak blue, 0.25 g). The sole blue and red lights promoted plants with higher dry weights under high light intensity (strong blue, 0.40 g, and strong red, 0.43 g), but plants under white light had the highest total dry weight (0.65 g), followed by plants grown under pink light (0.54 g).

We estimated the amount (mg) of each macro- (Figure 6) and micronutrient (Figure 7), multiplying the total dry weight by the concentration of nutrients in the leaves. Besides the lower concentration of N, K, and Cu (Figure 3), the multichromatic treatments (white and pink light) promoted higher nutrient uptake (mg plant^−1^), as shown in Figure 6 and Figure 7, because of the higher carbon fixation (dry weight), when compared to the other treatments.

## 3. Discussion

The effect of light intensity and light quality on plant growth and development is genotype-dependent [1], and photomorphogenesis is controlled by multiple photoreceptors of blue light, such as cryptochromes (CRYs, CRY1, CRY2), and of red/far-red light, such as phytochromes (PHYs, phy A to phy E) [24], that work together under blue and red lights [21]. As the light intensity affects the accumulation of biomass [13], it is necessary to adjust the light intensity to each plant material for optimum biomass accumulation [25].

We found the highest chlorophyll content in plants under blue light at high intensity and the lowest chlorophyll content in plants under red light at low intensity, but they did not differ from plants under strong red and weak blue light. Our results make clear that the intensity of blue light affects the chlorophyll content, because plants under strong blue light showed 22% more chlorophyll (24.37) than those under weak blue light (19.96). Regarding red light, the same did not happen, and both treatments resulted in plants with similar chlorophyll content. The chlorophyll content of plants under strong blue light did not differ from plants under the two multichromatic treatments (white light, 23.52; pink light, 22.28). Another study already observed that plants under low light intensity have decreased chlorophyll content, stomatal conductivity, and photosynthesis-related enzyme activity [26]. Other studies found that light at low intensity resulted in plants with reduced leaf widths [27] and thinner leaf blades, which could have led to a decrease in chloroplast number [28] and lower enzyme activity during the carbon cycle [29]. In contrast, another study [5] found the highest chlorophyll b content in plants with sole red light, and the lowest in plants with sole blue light.

We found etiolated plants cultivated under weak blue light and the smallest plants under white light, which did not differ from those under strong red, strong blue, and pink light. This shows that both intensity and quality affect plant etiolation. Plant height under weak blue light was 84% higher than that under strong blue light. This also shows that etiolation is not promoted by only blue or red photoreceptors. It does not depend only on light quality, but on its intensity too, and both photoreceptors work together [21]. Comparable results were found by [5], where soybean grown under sole red light was etiolated, and the plant height decreased (four-fold) with the increase in blue light up to the treatment of 80B:20R%, reaching appropriate plant heights under 25B:75R%, 50B:50R%, and 80B:20R%. According to the authors, the change in stem extension could not be attributed to a phytochrome-mediated response. According to [30], cryptochromes are suggested to have an inhibitory effect on hypocotyl elongation. Regarding the red light at the two tested intensities at the present study, the plant height did not differ.

Some authors have reported that using blue and red light simultaneously would suppress plant elongation more than that caused by using sole blue light or sole red light [15,19], but we can observe in the present study that strong blue, strong red, and pink light promoted similar plant heights (18.03, 18.14, and 17.88 cm, respectively). This could be an indication that wavelength sometimes can have a stronger effect than light intensity. In plants, investment in height improves access to light. It has been suggested that plants adapt phenotypically to different conditions of light and nutrient supply, supposedly to achieve the colimitation of these resources [5], and it can be seen that the most intense lights made the smallest plants, under strong red, strong blue, white and pink, which did not differ statistically.

The same light (white) that promoted the smallest plants in the present study resulted in the largest roots (28 cm), and the shortest roots were found in plants under strong blue and strong red lights (11.8 cm and 15.6 cm, respectively).

Another study [1] found that light intensity did not exert influence on root length, root frequency, fresh weight, dry weight, number of leaves per plant, leaf expansion, leaf area, and stomatal density in two cultivars of *Bixa orellana*, but they observed differences in chlorophylls and carotenoids. They also stated that the effect of light quality is genotype-dependent. In this study, root length did not differ statistically between the two studied red light sources, but plants under weak blue light showed roots that were 54% bigger than plants under strong blue light. Once again, we can see that the blue light effect is intensity-dependent.

In the present study, the lowest averages of total dry weight (Figure 5) were found under the low-intensity lights (weak red and weak blue), but they did not differ from the same wavelengths at high intensity, and plants under white light had the highest dry weight, followed by plants grown under pink light. This indicates that the multichromatic light was more effective for plant growth and to CO_2_ fixation. Plants treated with white light showed dry weights 148% higher than plants under weak blue light. Another important observation is that plants under white light had dry weights 86% higher than those under strong blue light and 67% higher than those under strong red light.

Another study showed that by increasing the multichromatic light intensity, the fresh weight of two cultivars of lettuce increased, and the maximum shoot fresh weight was observed under the strongest LED light (300 μmol m^−2^ s^−1^), and it was 306.9% higher than that observed in a greenhouse during the winter [27]. This is explained because the net photosynthesis increased with increasing multichromatic light intensity [13,27]. In vitro plantlets of *Bixa orellana* under fluorescent light had higher dry weights (0.14 g) compared to those under white LEDs (0.09 g), while the stomatal density was higher under white LEDs (140 stomata mm^−2^) when compared to fluorescent light (90 stomata mm^−2^) [1].

Plants under pink light had the highest stomatal density (Figure 3) and did not differ from those under strong blue and white lights. These are the treatments that promoted the highest dry weights (Figure 6). However, this contrasts with previous reports, where plants showing vigorous growth had larger sizes and a smaller number of stomata compared to those showing retarded growth [10,31,32,33]. Plants under pink light showed 152% more stomata when compared to plants under weak blue light. We considered whether monochromatic light (sole blue) is not sufficient for a plant to have higher stomatal density, but we could see that plants under strong blue light had 94% more stomata than those under weak blue light. This makes it clear that besides the light quality, the intensity changes plant morphology. A recent study found that blue light increases stomatal density [34] because the blue light activation of cryptochromes promotes stomatal development because of the regulation of transcriptional factors [35]. Another study with *Withania* plantlets showed that monochromatic lights inhibit stomatal formation [36], as we can see in the present work with strong red and weak blue light, but strong blue light made 94% more stomata than weak blue light, showing that light intensity is important on stomatal differentiation, even in monochromatic light.

Plants under weak blue light had the highest concentrations of N and Cu (56% and 100% more than the result obtained in white light, respectively), while plants under white and pink lights had the lowest concentrations of N, and those under white, pink, and strong red lights had the lowest concentrations of Cu (Figure 3). This is because these last cited treatments resulted in the largest dry weights, which correlated with the estimated total amount of each macro- and micronutrient absorbed by plants (Figure 6 and Figure 7). In addition to the lower concentrations of N, K, and Cu (Figure 3), the multichromatic treatments (white and pink light) promoted higher nutrient uptake (Figure 6 and Figure 7), because of the higher carbon fixation (dry weight), when compared to the other treatments. Different conclusions were found in other studies, where pink lights (red/blue = 3:1) increased the concentrations of N, P, K, and Mg in lettuce plants, and sole blue light decreased the N content [37].

We observed that white was one of the treatments that resulted in the lowest leaf concentrations of Ca and the highest total dry weight. This can be explained because higher photosynthetic rates [27] and rapid plant growth under high multichromatic light intensities [38] can make plants unable to absorb sufficient calcium and translocate it to younger leaves [39].

These results make it clear that plant photomorphogenesis is controlled not only by wavelength, but also by light intensity. The effect of blue light on plant morphology is intensity-dependent. White light resulted in the smallest plants with the largest roots, some of the highest chlorophyll contents and some of the highest stomatal densities, and consequently, the highest dry weight.

## 4. Material and Methods

### 4.1. Plant Material and Seeding

Two experiments were installed in a growth room under controlled environmental conditions of 27 ± 2 °C and a 16 h photoperiod. Seeds of common bean (*Phaseolus vulgaris* L. cultivar BA0958) were placed in plastic trays, with areas of 0.135 m^2^, containing the commercial substrate Promix BX growing medium mycorrhizae—general purpose, which was previously autoclaved (at 121 °C and 20 psi for 20 min) to eliminate any microbial effects.

The irrigation was performed in another tray, placed below the plant tray. The fertilization was the equivalent of 450 kg ha^−1^ of NPK fertilizer 20-20-20, resulting in 90 kg of N, 90 kg of P_2_O_5_, and 90 kg of K_2_O ha^−1^. We also used 50 mL of Major MS [40] solution and 1 mL of Minor MS solution in each water tray (0.135 m^2^) to add all macro- and microelements.

### 4.2. Light Sources

Six different light sources were evaluated, as shown in Figure 8: strong blue—blue LED at high light intensity with PPFD of 308.093 μmol m^−2^ s^−1^, (with peak of 11.729 μmol m^−2^ s^−1^ at 449 nm wavelength); weak blue—blue LED at low light intensity with PPFD of 7.366 μmol m^−2^ s^−1^, (with peak of 0.161 μmol m^−2^ s^−1^ at 459 nm wavelength); strong red—red LED at high light intensity with PPFD of 206.861 μmol m^−2^ s^−1^, (with peak of 7.903 μmol m^−2^ s^−1^ at 632 nm wavelength); weak red—red LED at low light intensity with PPFD of 10.525 μmol m^−2^ s^−1^, (with peak of 0.325 μmol m^−2^ s^−1^ at 629 nm wavelength); pink—combined red + blue LED with PPFD of 109.782 μmol m^−2^ s^−1^, with peak at 667 nm (3.386 μmol m^−2^ s^−1^) and at 470 nm (0.197 μmol m^−2^ s^−1^); white—combined red + white LED with PPFD of 129.038 μmol m^−2^ s^−1^, with peak at 660 nm (3.195 μmol m^−2^ s^−1^) and at 448 nm (0.445 μmol m^−2^ s^−1^). All the light sources were placed 40 cm from the plant trays except strong blue and strong red, which were placed 100 cm from the plant trays, because of the smaller area that would have been reached if placed 40 cm from the tray. The spectra were measured using an Li-180 Spectrometer (Li-COR Inc., Lincoln, NE, USA) 20 cm above the plant trays.

### 4.3. Plant Growth and Measurements

In the first experiment, 41 days after sowing, we collected the uppermost mature leaf blade of each plant for the leaf analysis, performed at A&L Greatlakes Laboratories, to evaluate the concentrations of N, S, K, Mg, Ca, B, Zn, Mn, and Fe. As this analysis was destructive, we had to start a new experiment at this time to evaluate the other parameters.

In the second experiment, we measured the relative chlorophyll content and the stomatal density 42 days after planting (DAP) as described below, and plant height (cm), root length (cm), shoot dry weight (g), root dry weight (g), and total dry weight (g) (shoot dry weight + root dry weight) at 80 DAP. Dry weight was determined by oven-drying plants at 70 °C until they reached a constant weight.

### 4.4. Relative Chlorophyll Content

The relative chlorophyll content was evaluated as SPAD values by placing the last expanded leaf of each plant, counted from the top downwards, in a portable SPAD-502 chlorophyll meter (SPAD-502, Minolta Co., Ltd., Tokyo, Japan). All plants were selected and evaluated within each plot, and the average values were reported.

### 4.5. Stomata Analysis

The methodology for this evaluation was based on another study [3]. The middle third portion of the last fully expanded leaf was cut into approximately 1 cm × 1 cm sections. Impressions of the leaves were obtained by placing leaf sections on top of a thin layer of super glue (Elmer’s Products, Inc., Westerville, OH, USA), spread over microscope glass slides, and removing the leaf after drying. Impressions were obtained for both the adaxial and abaxial surfaces of the leaves. Stomata observations were performed under an optical Leica DMLB microscope (Leica microsystems, Buffalo, NY, USA) at 40× magnification. Images were recorded using a SPOT 4.7 digital camera coupled to the microscope and analyzed using the SPOT basic software (SPOT Imaging 5.6, Diagnostic Instruments, Inc., Sterling Heights, MI, USA). The number of stomata was counted under the microscope for both the abaxial and adaxial surfaces of leaves. One area of 5 mm in diameter was selected for observations for each plot. The number of stomata in each image corresponded to an area of 0.2 mm^2^, completed 42 days after planting.

### 4.6. Experimental Design and Statistical Analysis

The experimental design was completely randomized and consisted of six treatments, with three replications per treatment in the first experiment and four replications in the second experiment. Each replication consisted of 9 plants in a tray. Data were collected and submitted to analysis of variance (ANOVA) using the Sisvar statistical analysis program [41], with means compared by Tukey’s test at the 5% level of significance.

## 5. Conclusions

Plant photomorphogenesis is controlled not only by wavelength, but also by light intensity. Etiolation was observed in bean plants under blue light at low intensity, but when the same wavelength had more intensity, the etiolation did not happen, and the plant height was the same as plants under multichromatic lights (pink and white light). The smallest plants showed the largest roots, some of the highest chlorophyll contents, and some of the highest stomatal densities, and consequently, the highest dry weight, under white LEDs, showing that the multichromatic light at high intensity resulted in better conditions for the plants in carbon fixation. The effect of blue light on plant morphology is intensity-dependent. Plants under multichromatic light tend to have lower concentrations of N, K, Mg, and Cu in their leaves, but the final amount of these nutrients absorbed is higher because of the higher dry weight of these plants. Plants under blue light at high intensity tend to have lower concentrations of N, Cu, B, and Zn when compared to the same wavelength at low intensity, and their dry weight is not different from plants grown under pink light. New studies are needed to understand how and on what occasions intense blue light can replace red light in plant physiology.

## Figures and Tables

**Figure 1 plants-13-00441-f001:**
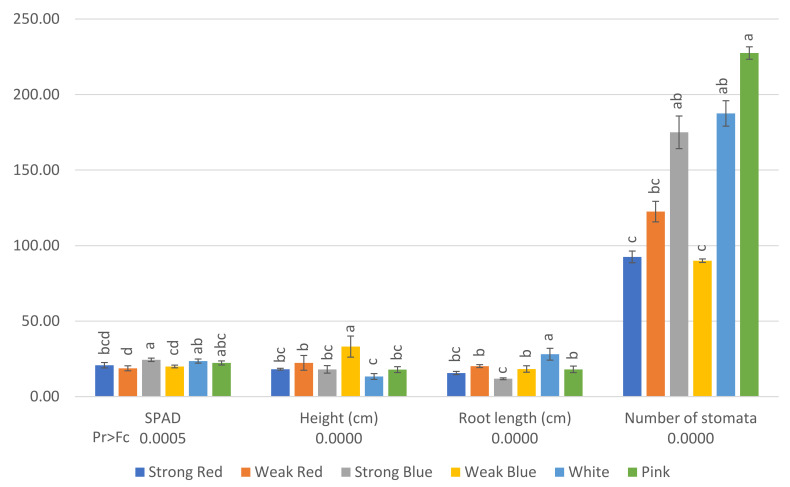
SPAD (relative chlorophyll content) and number of stomata (per square millimeter) on the abaxial surface of mature leaves of common bean 42 days after planting and plant height (cm) and root length (cm) 80 days after planting, under six light treatments. Bars with the same letters are not significantly different at the level of 1% of significance. The error bars represent the standard deviation.

**Figure 2 plants-13-00441-f002:**
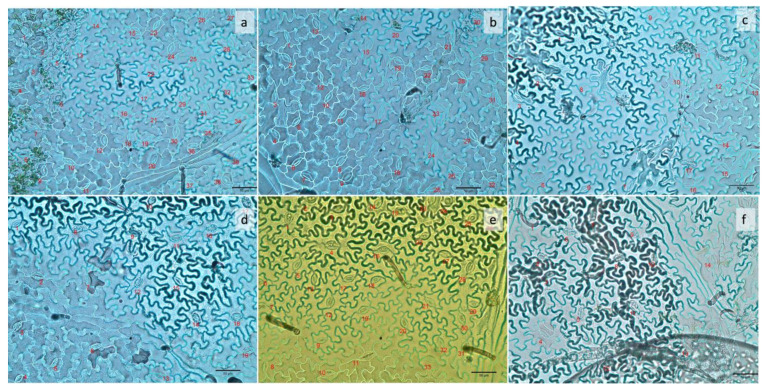
Number of stomata in the abaxial surface of common bean leaves 42 days after planting, under six different light treatments: (**a**) pink; (**b**) strong blue; (**c**) strong red; (**d**) weak blue; (**e**) white; (**f**) weak red. The number of stomata in each image corresponds to an area of 0.2 mm^2^.

**Figure 3 plants-13-00441-f003:**
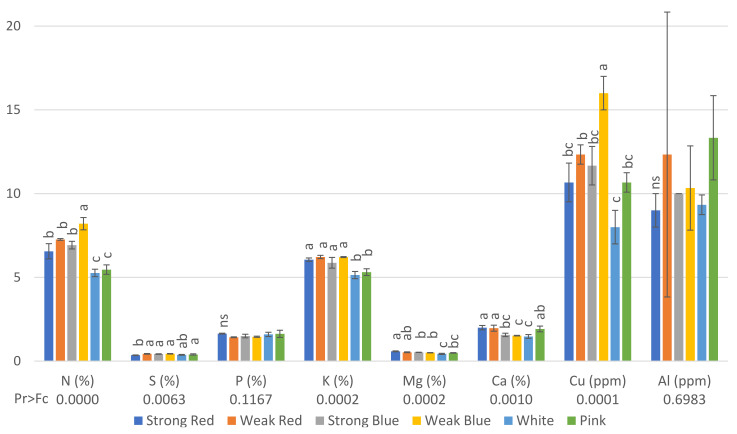
Concentration of N (%), S (%), P (%), K (%), Mg (%), Ca (%), Cu (ppm), and Al (ppm) on common bean (*Phaseolus vulgaris*) plants under six light treatments 41 days after planting. Bars with the same letters are not significantly different at the level of 1% of significance. ns means non-significant at 1% of probability. The error bars represent the standard deviation.

**Figure 4 plants-13-00441-f004:**
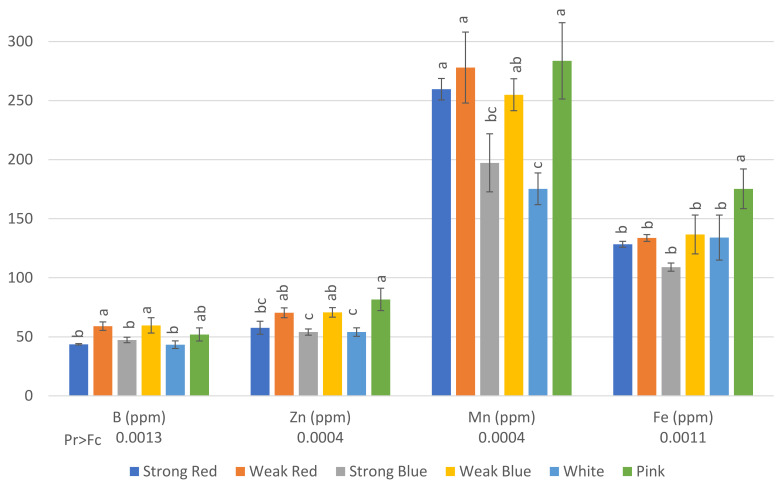
Concentration in ppm of B, Zn, Mn, and Fe on common bean (*Phaseolus vulgaris*) plants under six light treatments 41 days after planting. Bars with the same letters are not significantly different at the level of 1% of significance. The error bars represent the standard deviation.

**Figure 5 plants-13-00441-f005:**
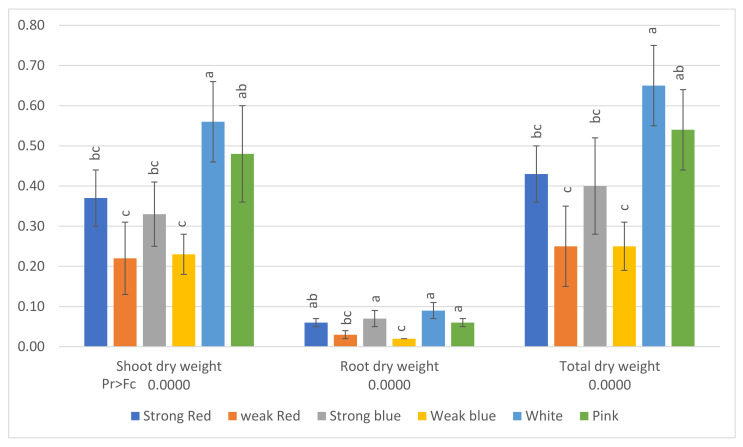
Shoot dry weight, root dry weight, and total dry weight (g) of common bean (*Phaseolus vulgaris*) plants under six light treatments. Bars with the same letters are not significantly different at the level of 1% of significance. The error bars represent the standard deviation.

**Figure 6 plants-13-00441-f006:**
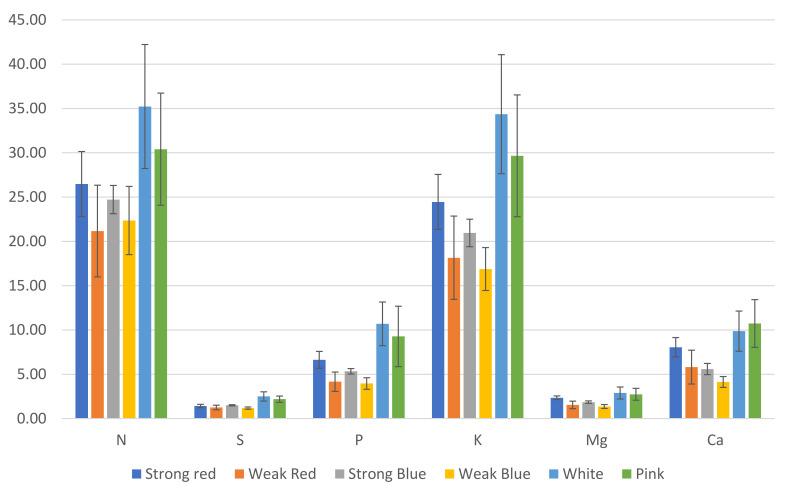
Estimated amount (mg) of macronutrients in common bean leaves, based on leaf concentration and total dry weight. The error bars represent the standard deviation.

**Figure 7 plants-13-00441-f007:**
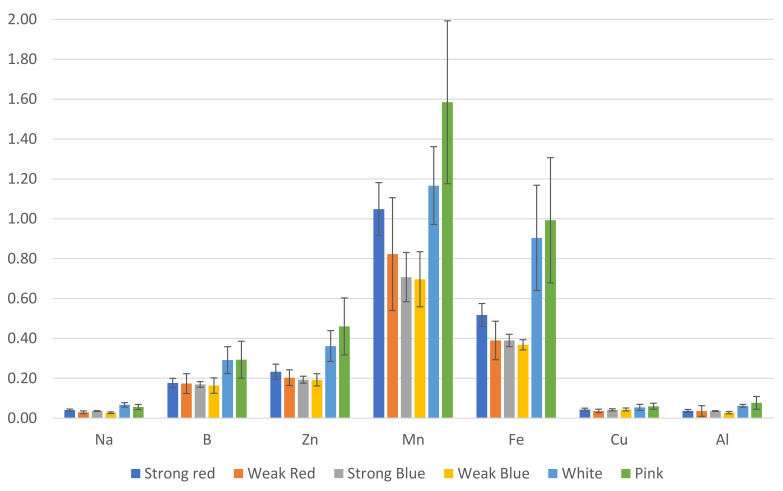
Estimated amount (mg) of micronutrients and sodium in common bean leaves, based on leaf concentration and total dry weight. The error bars represent the standard deviation.

**Figure 8 plants-13-00441-f008:**
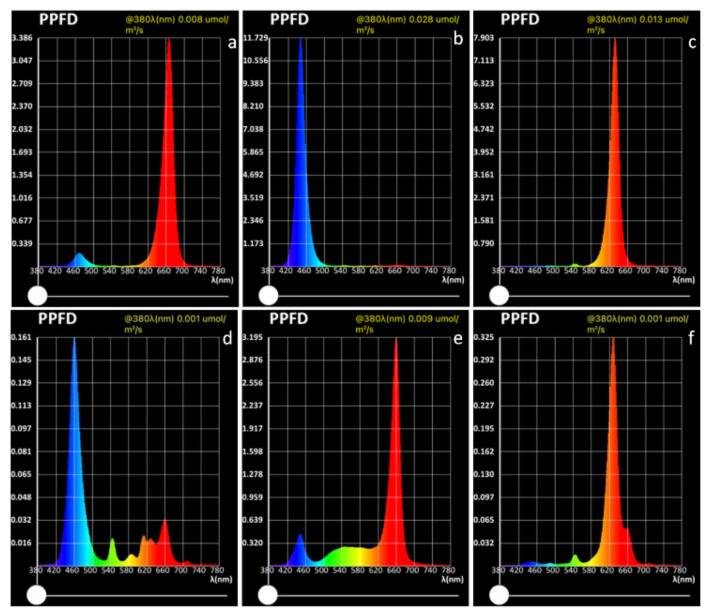
Intensity and composition of lighting as photosynthetic photon flux density (PPFD) for the different light treatments in this study: (**a**) pink; (**b**) strong blue; (**c**) strong red; (**d**) weak blue; (**e**) white; (**f**) weak red.

## Data Availability

Data available upon request.

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
