# Peer review of "Wavelength and Light Intensity Affect Macro- and Micronutrient Uptake, Stomata Number, and Plant Morphology of Common Bean (Phaseolus vulgaris L.)"

_plants, 2024, doi:10.3390/plants13030441_

Round 1

Reviewer 1 Report

Comments and Suggestions for Authors

The submitted manuscript to PLANTS-MDPI entitled “Wavelength and Light Intensity Affect Macro and Micronutrient Uptake, Stomata Number and Plant Morphology of Common Bean (Phaseolus vulgaris L.)” is interesting to investigate. BUT, following are the comments that need to be addressed:

Line 14-23: The methods and treatments should be explained in the M*M section. In abstract, authors should briefly mention the treatment details.

The keywords are not appropriate.

The abstract section is poorly arranged. The results are not so convincingly presented, the conclusion and future directions are not mentioned either.

 Authors did not mention the research gap in the second last paragraph of introduction section.

Please always mention the statistical significance letters throughout the MS.

The presentation of results is not clear. Please remove the inverted commas and present the results with fold changes or percentage changes to make them better understandable.

The paragraphs in the discussion section are very small. Please merge them accordingly.

The conclusion section should also be in one paragraph.

Line 320: How can you propose that even though you do not have enough data for these two processes?

Apart from them, there should be more physiological data to support the conclusion section.

Author Response

Dear reviewer 1,

We thank you for your review and your comments in our work. It helped us to see some details that we have missed.

We answered your comments and improved the manuscript according to your comments, please see the attachment. Everything that was corrected or rewritten is now in red font. Please see the comments throughout the manuscript.

Best regards,

Bueno and Vendrame

Reviewer 2 Report

Comments and Suggestions for Authors

This study evaluated plant growth, stomata formation, chlorophyll index, and absorption of macro and micronutrients by common bean plants under six light treatments. Etiolation was observed in bean plants under red and blue lights at low light intensity. Under the same sole wavelengths but with higher intensity, the etiolation was much lower, with the same shown by plants under blue+red light. The smallest plants showed the largest roots, the highest dry weight and one of the highest stomatal densities, under white LED. The highest chlorophyll content was observed in plants under “strong blue” light and the lowest under “weak red”, unlike other reports, that found the highest chlorophyll content in plants under sole red light, and the lowest under sole blue light.

There are many issues that need to be addressed with this first version of the manuscript before it can be considered for publication in this or another journal. Here are my line-by-line comments:

Line 12: after about add: "the effect of", otherwise the phrase makes no sense

Line 21: Why 0.2 mm2 ? Where is this coming from? Why not report stomatal density per 1 mm2 ?

Line 22: that "e" should be "and" 

Line 36 and the rest of the manuscript: author names and year is not the in-text citation style used by THIS journal. Use sequential numbering of references and cite by numbers in square brackets.

Line 94: abbreviations R and B should be defined here, at the location where they are first encountered

Line 108: Details for growth room needed (manufacturer, size/dimensions, control system for temperature and for light, distance from light source to plants)

Line 112: Details needed about autoclave and about the settings used

Line 118: Indicate vendors for all light sources used

Line 134: Details about this analysis needed: method used, apparatus, standards used for calibration, calibration method etc.

Figures 2-5: What does "Pr>Fc" mean? Explain this in your manuscript!

Figure 3 caption: you are not showing just numbers in this figure, but optical micrographs. Figure caption should be rephrased.

Figure 4: fonts too small, impossible to read both axes

Figure 4: How come some concentrations are expressed as absolute values (in ppm) and others are expressed relatively (as %)? Either express uniformly or at least explain the latter, % out of what?

Line 187-188: if it did not differ significantly that you should put some identical markings on the corresponding bars in the figures, and different markings where there are significant differences. For example identical letters and different letters (aaabb...) This should be done for ALL figures.

Line 204: How was this "estimation" performed? Provide all details!

Line 328: Please speculate with respect to the possible causes of these conflicting findings.

References should be listed not alphabetically, but in the order of their citation in the text and formatted according to this journal's requirements. Also, I find the number of references too small, given that there is a lot of literature on this topic published.

Comments on the Quality of English Language

Minor edits needed here and there. No major issues.

Author Response

Dear reviewer 2,

We thank you for your review and your comments in our work. It helped us to see some details that we have missed.

We answered your comments and improved the manuscript according to your comments, please see the attachment. Everything that was corrected or rewritten is now in red font. Please see the comments throughout the manuscript.

Best regards,

Bueno and Vendrame

Round 2

Reviewer 1 Report

Comments and Suggestions for Authors

Regarding figure 6 and 7, the author sated that "Differences can be seen using the standard deviations (error bars)."

However, in my opinion the error bars only provide the difference between the replicates of the respective treatments. To see the difference between treatments, you must do variance analysis. 

Otherwise, the manuscript can be "ACCEPTED" for publication. 

Congratulations to authors for their significant contribution to the science. 

Reviewer 2 Report

Comments and Suggestions for Authors

I have no further comments